# Development of a small and sick newborn clinical audit tool and its implementation guide using a human-centred design approach newborn clinical audit process and design

Muthoni Ogola[1,2,3]*, John Wainaina[1], Naomi Muinga[1], Wairimu Kimani[4], Maureen Muriithi[3], Jalemba Aluvaala[1,2,5], Mike English[1,5], Grace Irimu[1,2]

1 Health Services Unit, KEMRI-Wellcome Trust Research Programme, Nairobi, Kenya, 2 Department of Paediatrics and Child Health, University of Nairobi, Nairobi, Kenya, 3 Pumwani Maternity Hospital, Nairobi, Kenya, 4 Kenyatta National Hospital, Nairobi, Kenya, 5 Centre for Tropical Medicine and Global Health, Nuffield Department of Medicine, University of Oxford, Oxford, United Kingdom

* mogola@kemri-wellcome.org

## Abstract

Clinical audits are an important intervention that enables health workers to reflect on their practice and identify and act on modifiable gaps in the care provided. To effectively audit the quality of care provided to the small and sick newborns, the clinical audit process must use a structured tool that comprehensively covers the continuum of newborn care from immediately after birth to the period of newborn unit care. The objective of the study was to co-design a newborn clinical audit tool that considered the key principles of a Human Centred Design approach. A three-step Human Centred Design approach was used that began by (1) understanding the context, the users and the available audit tools through literature, focus group discussions and a consensus meeting that was used to develop a prototype audit tool and its implementation guide, (2) the prototype audit tool was taken through several cycles of reviewing with users on real cases in a high volume newborn unit and refining it based on their feedback, and (3) the final prototype tool and the implementation guide were then tested in two high volume newborn units to determine their usability. Several cycles of evaluation and redesigning of the prototype audit tool revealed that the users preferred a comprehensive tool that catered to human factors such as reduced free text for ease of filling, length of the tool, and aesthetics. Identified facilitators and barriers influencing the newborn clinical audit in Kenyan public hospitals informed the design of an implementation guide that builds on the strengths and overcomes the barriers. We adopted a Human Centred Design approach to developing a newborn clinical audit tool and an implementation guide that we believe are comprehensive and consider the characteristics of the context of use and the user requirements.

**Data Availability Statement:** The data for the study are within the article and its Supporting Information files.

**Funding:** This work was supported by a Senior Research Fellowship awarded to The Wellcome Trust (#207522 to ME) and by a grant from the Initiative to Develop African Research Leaders (IDeAL) through the DELTAS Africa Initiative (DEL-15-003), an independent funding scheme of the African Academy of Sciences (AAS) 's Alliance for Accelerating Excellence in Science in Africa (AESA) and supported by the New Partnership for Africa's Development Planning and Coordinating Agency (NEPAD Agency) with funding from the Wellcome Trust (107769/Z/10/Z) and the U.K. government awarded to ME. For Open Access, the author has applied a CC-BY public copyright license to any author accepted manuscript version arising from this submission. The funders had no role in the preparation of this report or the decision to submit it for publication.

**Competing interests:** The authors have declared that no competing interests exist.

## Introduction

Newborn mortality remains a burden in sub-Saharan Africa (SSA) with the majority (82%) of the deaths occurring due to conditions that can be prevented using low-cost high impact interventions [1]. The fact that the most common causes of newborn deaths in SSA are preventable means that health care workers can learn lessons from these deaths and improve newborn quality of care [2]. Clinical audits are one way through which health care workers can identify modifiable factors in care and improve the quality of care by acting on them [2, 3]. Modifiable factors are factors that may have prevented the occurrence of an adverse event if done differently [2].

In several countries, the maternal and perinatal death review process has improved the quality of care and, subsequently, reduced maternal and perinatal mortality [4–7]. The maternal and perinatal death review process implemented in many low and middle-income countries (LMICs) is referred to as the Maternal and Perinatal Death Surveillance and Response (MPDSR) [8, 9]. The MPDSR tool is a structured guide used in the collection of detailed information on the care provided to the mother during pregnancy, labour and delivery and the immediate resuscitation provided to the newborn after birth [10]. The perinatal period, as defined by the WHO, "commences at 22 completed weeks of gestation and ends at seven completed days after birth" [11]. The neonatal period commences at birth and ends at 28 completed days after birth [7]. To diligently audit the quality of care provided to the newborn, in addition to the immediate resuscitation after birth, attention must also be paid to the post-resuscitation care as well as care in the newborn unit [12].

We previously conducted a scoping review to identify the modifiable factors in the care of the small and sick newborns (SSNBs) and to assess the quality of perinatal and newborn clinical audits in LMICs. One of the determinants of a quality audit process was the availability of a structured audit tool that covered three periods of newborn care which are: (i) immediate newborn care and resuscitation after birth, (ii) post resuscitation care, and (iii) care in the newborn unit. However, our scoping review, identified a dearth of perinatal and newborn care clinical audit tools that are specifically designed to audit the care of SSNBs beyond the initial resuscitation after birth [12].

We, therefore, designed a tool to audit the quality of care provided to the SSNBs who died or who survived but had a near miss event [12]. This tool would complement the MPDSR by allowing for the review of SSNB care through the three periods of care. We also designed its implementation guide. Quality design begins by trying to understand where the real gaps in the current newborn audit process are and therefore solving the right problem and doing so in a way that meets human needs and capabilities [13]. It is therefore important that the design takes into consideration the usability and human factor characteristics of the audit tool, and hence the experience it provides for the user [14, 15]. This can be achieved through the use of a Human-Centred Design (HCD) approach. This "is a design approach that puts human needs, capabilities, and behaviour first, then designs to accommodate those needs, capabilities, and ways of behaving" [13]. This promotes the development of usable systems by advocating for active user participation and allowing for several iterations of a design of the system and subsequent modifications based on the users' requirements [16, 17]. A HCD approach was considered appropriate for the design of a context sensitive clinical audit tool and implementation guide as it presumes that the users of an innovation understand its core challenges, and therefore hold the key to its solution [18].

The objective of the study was to co-design a comprehensive SSNB clinical audit tool and its implementation guide that takes into consideration the basic needs, capabilities and limitations of the health workers who will be the end-users while taking into account the key principles of HCD (Box 1).

---

**Box 1. Key principles of the human centred design process**

1. Understanding the users, their environment and how these influence use of the audit process.

2. Active user participation throughout the development, analysis and evaluation of the audit process.

3. Early prototyping to develop design solutions based on a shared understanding of the users and their requirements.

4. Continuous iteration of the design solutions. This includes a cyclic process of designing the prototype tool, evaluating its usability on real tasks with the users and redesigning based on user feedback.

5. Use of multidisciplinary design teams.

## Methods

The methods section is divided into three phases; phase one describes the design of the SSNB audit tool; phase two describes the design of the implementation guide, and phase three describes the usability testing of the two tools.

### Study sites and participants

This study was conducted in hospitals participating in a clinical information network (CIN) for neonates. CIN-Neonatal is a collaborative effort between the Ministry of Health (MoH), researchers and the participating hospitals that aims to improve the quality and utilisation of patient-level data [19]. CIN-Neonatal consists of 21 purposively selected public hospitals from 12 out of the 47 Counties in Kenya [20]. Each facility NBU has a focal paediatrician/s, a nurse in-charge of NBU and the health records information officer (HRIO) who act as a link between the hospital and the research teams [21]. Newborn deaths in these hospitals contribute to two-thirds of mortality among patients aged 0–13 years with five conditions accounting for the majority of newborn admissions; Intrapartum related conditions, respiratory distress syndrome, neonatal sepsis, neonatal jaundice and low birthweight [20].

**Description of study sites for the design of the audit tool.** Pumwani Maternity Hospital (PMH) is one of the County referral hospitals in Kenya and was selected as the site for the design of the SSNB audit tool [22]. This hospital was selected as at the time of the study, the lead researcher (MO) had been a paediatrician in the hospital for approximately one year. It is also the largest referral maternity hospital in Kenya located in Nairobi County with approximately 100 deliveries per day [23]. The NBU admits approximately 350–400 newborns per month and provides intermediate-level care [24]. The hospital held monthly MPDSR meetings to discuss maternal near misses with a brief overview of perinatal morbidity and mortality statistics. In the event of a maternal death, the meeting was held within 48 hours of the death.

**Description of study sites for usability testing.** The initial plan was to test the tools in PMH. However, Kenyatta National Hospital (KNH), the largest teaching and referral hospital in Kenya expressed interest in improving its newborn clinical audit process and was

incorporated into the study as a testing site in January 2021. KNH caters for low-and-middle-income populations from Nairobi and its environs as well as referrals from other hospitals in the country and the greater Eastern Africa region [25]. The NBU of KNH admits approximately 250–300 newborns per month and offers intensive care to critically ill newborns [24]. The NBU staff do not attend MPDSR meetings, they however infrequently hold in-house meetings to discuss the NBU morbidity and mortality statistics.

## Study participants

**Cognitive walkthrough and usability testing of prototype audit tools in the hospitals.** The SSNB audit tool was intended for use by a transdisciplinary team who directly provide care to the SSNB and were therefore involved in its design and testing its usability (Box 2).

---

Box 2. Study participants

Study participants–Cognitive walkthrough of the prototype audit tools and usability testing of the audit tool and implementation guide

a  Paediatricians.

b  NBU nurses.

c  Neonatology fellows (Paediatricians enrolled in a two-year fellowship programme to specialize in neonatology) from the University of Nairobi (UoN).

d  Paediatric residents (trainee paediatricians enrolled in a three-year postgraduate training programme in Paediatrics and Child Health) from the UoN.

e  Junior clinicians (medical officer interns, medical officers, clinical officer interns and clinical officers) from the County hospital NBUs.

f  Other cadres participating in newborn care (hospital administrators, nutritionists, representatives from laboratory, pharmacy, records, biomedical, occupational and physiotherapy departments).

---

**Virtual design workshop.** The design workshop of the implementation guide involved 37 purposely selected participants (two neonatologists, 17 NBU nurse leaders, 17 paediatricians and an MoH official) and was held using the Zoom platform.

## Development and iterative testing of the audit tool and implementation guide

We describe the methods used for the development of the audit tool and its implementation guide in three phases: (i) Development of draft zero of the newborn clinical audit tool by the research team and refining it using cognitive walkthrough methodology, (ii) development of an implementation guide adapted from the Geneva: World Health Organisation; 2018 manual through consensus opinion of the end users and iii) usability testing of the audit tool and implementation guide.

A modified version of a three step HCD approach was used across all three phases in the design of the audit tool and its implementation guide [14]. Briefly, the three steps in HCD applied in our study included:

1. Understanding the context of use, user requirements and understanding the structure of the available audit tools. This led to the development of the implementation guide and draft zero of the prototype of a SSNB audit tool [2, 3, 10, 26].

2. Cognitive walkthrough which refers to a structured approach assessing the usability of the prototype audit tool and identifying barriers to its use. The outcome was a high-level prototype audit tool which was ready for testing with the end users.

3. Usability testing which refers to the process of field testing the feasibility of the audit tool and implementation guide as the standard operating procedure (SOP) by which newborn audits would be conducted in the Kenyan public hospitals. The outcome of this was an audit tool and implementation guide that are scalable.

The audit tool and implementation guide design process and timelines for each phase are described in Fig 1 below.

## Phase one—Development of draft zero of the audit tool and further modification by end-users using cognitive walkthrough methodology

**a). Design of draft zero of the prototype audit tool.** We studied the existing audit tools to inform the structure of draft zero of the SSNB prototype audit tool. These audit tools included: Kenyan MoH MPDSR Tool, WHO Stillbirth and Neonatal Death Case Review Form and WHO Child and Neonatal Death Review Form [2, 3, 10]. We also made reference to the modifiable factors in newborn care identified from a scoping review, Kenyan Basic Paediatric Protocols, Comprehensive Newborn Care Protocols and WHO guidelines that outline standards of care for the SSNB [12, 21, 27–30]. The outcome of this initial step was draft zero of the audit tool which was paper based.

**b). Cognitive walkthrough of draft zero and subsequent modified prototypes of the audit tool.** The newborn unit audit meetings were initially held monthly then two-weekly.

Draft zero of the audit tool was subjected to a five-step iterative process during NBU audit meetings in PMH. The iteration involved the sequence of developing a prototype tool, evaluating it with the end users and modifying it based on their feedback to improve its efficiency and ensured that it could be used by the intended users with minimal coaching (Box 3). The tool went through 15 revisions during the study period until we reached a point of saturation on the feedback from the end users during each audit cycle. The outcome of this was a high-quality prototype audit tool.

## Phase two—Development of a context sensitive audit implementation guide by obtaining consensus opinion of the end users

**a. Understanding the context through a review of literature.** Initial work involved understanding the context of use and user requirements. We reviewed literature that described the Kenyan health system context in terms of: (i)The organizational environment which include organizational culture, values, leadership [20, 31–33], (ii) physical environment with respect to structures, availability of equipment, medicines and materials [31, 34], (iii) health workers who are the end users in terms of their behaviours, attitudes and work tasks [34–37], and (iv) literature from other LMICs that described the facilitators and barriers to the maternal and perinatal audit process [38–45]. This information was used to develop the topic guide for the focus group discussions.

**b. Understanding the context through focus group discussions and arriving at consensus on the audit implementation guide.** To further understand the context and user

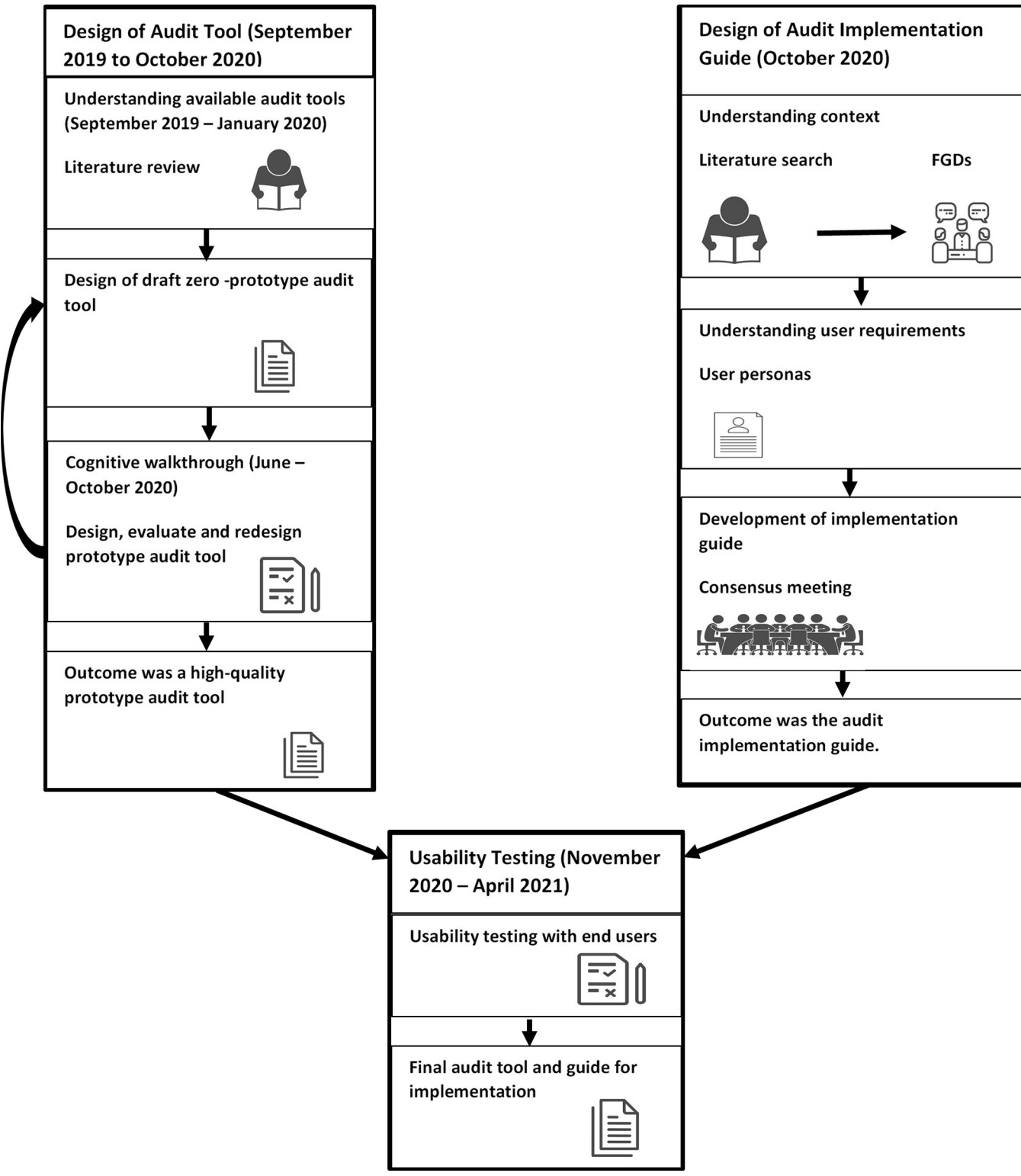

**Fig 1. Human centred design approach of a small and sick newborn clinical audit tool and implementation guide.**

Box 3. Five step process in the cognitive walkthrough methodology

1. **Step one** involved the selection of the team that would prepare the case summary. The researcher identified the team (junior clinician and nurse) responsible for preparing for and presenting in the audit meeting. They would agree on a suitable day and time to meet and begin the selection of a case and collecting information on the case to be audited.

2. **Step two** began with selecting a case for the audit. This involved the lead researcher (MO) discussing with the paediatricians and NBU nurse leader and selecting a diagnosis of interest based on an area they felt needed to be highlighted. We would then settle on one case that fit an agreed upon criteria. The research team would walk the audit team chosen in step 1 through the process of summarising the case onto the audit tool and after every section would receive feedback from the audit team on identified new and unmet requirements of the audit tool based on their experience while interacting with it. The feedback included i) the structure of the audit tool to ensure the systematic flow of information ii) missing information from the audit tool that was important in newborn management and iii) information that was not considered mandatory for the audit tool.

3. **Step three** involved the modification of the audit tool based on the experience of the user in step two. The documented feedback was discussed with the research team and modifications to the audit tool made based on this evaluation before the audit meeting.

4. **Step four** involved seeking feedback from a broader audit team based on experience of using the audit tool. This took place during the audit meeting. Each participant attending the audit meeting was provided with the modified audit tool and encouraged to fill in the tool as the audit case was presented. This ensured that a wider proportion of end users experienced using the audit tool and therefore giving diverse input into its design. After the meeting, the lead researcher (MO) would invite willing participants for a debrief session to give feedback on their thoughts and experience of using the tool and suggest the areas that required modification.

5. **Step five** involved revision of the audit tools based on the broader team experience in their use. This step involved the researcher discussing with the research team the feedback from audit meeting participants and making modifications to the audit tool in preparation for the next audit cycle.

requirements, we conducted a consensus virtual workshop to engage the end users in the design of the audit implementation guide. The workshop consisted of three parts that included: (i) FGDs that allowed for in depth understanding of the facilitators and barriers to the newborn audit process within the Kenyan context (described in detail in S1 Appendix), (ii) a plenary session to discuss the audit implementation guide based on WHO recommendations that ensured all participants had the same understanding of an ideal implementation guide, and iii) a consensus meeting to agree on feasible SOPs for the implementation guide that would adapt the Geneva: World Health Organisation; 2018 manual with the context and users in mind [2]. The consensus meeting began with designing user personas to understand the

attributes of the end users before arriving at consensus using the nominal group technique. The users were involved in every step of the design with a full description of the workshop provided in S1 Appendix. The outcome of which was the SSNB audit implementation guide.

## Phase three—Usability testing of the newborn clinical audit tool and implementation guide

The usability testing of the audit tool and implementation guide involved the lead researcher (MO) taking a facilitative role while allowing the hospital teams to take over the management of the newborn audit process following the recommendations in the implementation guide. The usability of the audit tool was also tested in PMH and KNH, a tertiary hospital, over a period of six months from November 2020 to April 2021. The audit meetings in the tertiary hospital were fully virtual as this was preferred by the hospital team due to the COVID-19 pandemic.

In PMH, a multidisciplinary audit team was constituted to manage the newborn audit process. This team was composed of eight members as recommended in the implementation guide with the paediatrician as the chair. This team employed the implementation guide recommendations.

At the start the researcher (MO) held a meeting with the audit committee where she described their roles as specified in the implementation guide and advised that they would be responsible for managing the audit process. MO was available to provide guidance where needed, make observations as well as receive feedback from the members of the committee on components of the implementation guide that were difficult to implement. These were documented and modified accordingly.

The audit tool was tested in the tertiary hospital as the health workers (residents, fellows and nurses) would prepare for and present in the audit meeting under the supervision of the chair of the audit committee. They would engage with the researcher at each stage to seek clarification as well as provide feedback on the usability of the tool. All necessary modifications were made to the audit tool before each meeting.

## Ethics

Ethical approval for this study was granted by the Kenya Medical Research Institute (KEMRI) Scientific and Ethics Review Unit (Protocol no: KEMRI/SERU/CGMRC/161/3852) with additional approval from Kenyatta National Hospital- University of Nairobi Ethics and Research Committee (KNH-UoN ERC (P330/06/2020)) for the testing phase. The participants signed an electronic consent form and emailed it to the lead researcher (MO) three days before the workshop (S2 Appendix). In addition, the participants gave consent again before the FGD by directly messaging the group moderator "I agree", there were however a few participants who were not proficient in using the Zoom platform and were allowed to unmute and give their verbal consent. We applied for a waiver of the consent process during the audit meetings. This was because the feedback and observations conducted and documentation of the same were on the audit tool and audit process (our experiences as facilitators during the audit meetings, where people were positioned in relationship to one another, people's behaviours etc.). We did not document any details of the cases discussed therefore ensuring confidentiality. Due to the sensitive nature of mortality audit meetings, obtaining individual informed consent from each participant during the audit meetings may have affected the quality of discussions taking place during the meeting as the participants may have feared that the study was focusing on the cases discussed which was not the case. This would therefore have negatively affected the qualitative results obtained from the study.

## Results

### Outcome of the co-design of the small and sick newborn clinical audit tool

**a. Outcome of reviewing the available audit tools to understand the gaps and inform the structure of draft zero of the SSNB audit tool.**   The review of the three maternal, perinatal and newborn audit tools informed the structure of draft zero of the prototype newborn audit tool. The Kenyan MPDSR tool and the WHO stillbirth and neonatal death case review form were focused on maternal and perinatal care. The WHO child and neonatal death review form had information on neonatal care but focused on care beyond the neonatal period [2, 3, 10].

The Kenyan MPDSR tool and the WHO stillbirth and neonatal death audit tool were similar in their structure. Both tools followed the same sequence to obtain patient information in a systematic manner. The sequence began with hospital details, biodata of mother and baby, mother's antenatal care, labour and delivery details, resuscitation care of baby, cause of death, modifiable factors and action plans. The WHO child and neonatal death review form did not entirely focus on the perinatal and neonatal period, however it had content on the care of children beyond the neonatal period. The structure of the audit tool however followed the same sequence of patient care details from admission to death with more provision to discuss the care provided during hospital stay. A summary of the structure and content of the three studied tools has been presented on Table 1 below.

**b. Structure of draft zero of the prototype newborn audit tool.**   Draft zero of the prototype newborn audit tool was divided into four sections (S3 Appendix):

1. Newborn biodata which included the patient admission number, gender, date of birth, birth weight, gestation at birth, age at death (mortality audits) or age at review (near miss clinical audits), weight at death (mortality audits) or weight at review (near miss audits).

2. Mother's details which included the ANC details and details of labour and delivery. ANC details included: Mother's blood group, Human Immunodeficiency Virus (HIV) status, syphilis status, hypertension in pregnancy and diabetes in pregnancy.
   Labour and delivery details included: Mode of delivery, complications during labour and delivery and if the newborn was resuscitated after delivery. The complications of labour and delivery were coded at the bottom of the audit tool.

3. Review of the care provided during and after admission.
   This section of the audit tool began by reviewing the care provided at admission and was structured to allow the audit participants to discuss the quality of care provided; what was well done, what could have been done differently and the recommendations made for each step in care during admission (timely admission, adequacy and appropriate assessment -history, physical examination, and investigations, primary and secondary diagnoses, supportive and definitive management).
   The next section reviewed the post admission supportive and definitive management based on the progression of illness.

4. Details of death for mortality audits (primary and secondary cause of death) and the conditions that led to unfavourable outcomes for near miss audits.

**c. Final outcome of the cognitive walkthrough of draft zero and the subsequent modified prototype audit tools.**   The feedback received during the cognitive walkthrough phase was categorised into one of the three groups:

1. Usability—the extent in which the content of the audit tool allowed the users to comprehensively summarise the care provided to the small and sick newborns.

**Table 1.  Summary of structure and content of available perinatal, neonatal and child clinical audit tools.**

| Basic design | Kenyan MOH MPDSR Tool [10] | WHO Stillbirth and Neonatal Death Case Review Form [3] | WHO Child and Neonatal Death Review Form [2] |
|---|---|---|---|
| 1. Structure of input fields | | | |
| • Closed ended questions designed for single word and yes/no responses | ✓ | ✗ | ✗ |
| • Closed ended questions with free text. | ✗ | ✓ | ✗ |
| • Open ended questions with free text | ✗ | ✗ | ✓ |
| 2. Details of facility where death occurred | ✓ | ✓ | ✓ |
| 3. Biodata of patient | ✓ | ✓ | ✓ |
| 4. Biodata of mother | ✓ | ✓ | ✗ |
| 5. If patient a referral in and if yes, details of facility referred from | ✗ | ✓ | ✓ |
| 6. ANC attendance | ✓ | ✓ | ✓ |
| 7. Obstetric conditions during pregnancy, labour and delivery | ✓ | ✓ | ✓ |
| 8. Management of labour and delivery | ✓ | ✓ | ✗ |
| **Newborn details** | | | |
| 9. Resuscitation of baby | ✓ | ✓ | ✗ |
| 10. Description of clinical illness and progression | ✗ | ✗ | ✓ |
| 11. Investigations done and key results | ✗ | ✗ | ✓ |
| 12. Primary and underlying diagnoses | ✗ | ✗ | ✓ |
| 13. Treatment provided | ✗ | ✗ | ✓ |
| 14. Cause of death | ✓ | ✓ | ✓ |
| 15. Modifiable factors | | | |
| • A list with a checkbox for selection. Categorised as the three delays. | ✓ | ✗ | ✗ |
| • A list that is categorised as family, administrative and provider related modifiable factors. | ✗ | ✓ | ✗ |
| • Free text section to list down identified modifiable factors. | ✗ | ✗ | ✓ |
| 16. Action plans | | | |
| • Free text section to list down action plans. | ✓ | ✓ | ✗ |
| • Structured action plan summary form separate from audit tool | ✗ | ✗ | ✓ |

Abbreviations: ANC, Antenatal care; MPDSR, Maternal and Perinatal Death Surveillance and Response; NBU, Newborn unit; WHO, World Health Organisation.

2. Human factors—the application of feedback on the structuring of the audit tool that enabled or limited the users' interaction with the audit tool.

3. User experience—the perceptions and thoughts of the users based on their experience of using the audit tool.

A detailed description of the results and outcome of the modifications made based on user feedback during each audit cycle have been summarised in S4 Appendix.

The difference between the co-designed SSNB clinical audit tool and the available audit tools are summarised in Box 4 below.

## Outcome of the development of an audit implementation guide adapted to the context

**a. Characteristics of the context and users that were perceived as facilitators and barriers to using the newborn clinical audit process based on the focus group discussions.**   The

**Box 4. Differences between the co-designed SSNB audit tool and the available audit tools**

| Design details | Structure of co-designed audit tool | Structure of available audit tools |
|---|---|---|
| **1. Sections of the audit tools.** | • The audit tool has two sections. The first section is to provide a summary of the care provided. The second section is a guide for documenting the discussion on the quality of care in each section. | • Only the WHO child and neonatal death review form has provision to document the gaps identified during the audit meeting discussion. The discussion however comes after each section describing the summary of care. |
| **2. Structure of input fields.** | • The basic structure mostly includes check boxes, drop-down calendars and open-ended questions. | • The MPDSR tool and the WHO stillbirth and neonatal death case review form have fields structured as closed-ended questions. The WHO child and neonatal death review form is structured as open-ended questions with free text.<br>None of the tools have checkboxes and drop-down calendars. |
| **3. Resuscitation and post-resuscitation care.** | • The section on newborn resuscitation immediately after delivery specifies the type of resuscitation provided and includes the post-resuscitation care provided. | • The available audit tools only document if the newborn was resuscitated but do not provide details of the resuscitation. They do not provide details on post-resuscitation care. |
| **4. Details of admission.** | • The details of admission include details on delays in transfer from the delivery unit to the NBU and delays in the review of the newborn by the clinician while in the NBU. | • The available audit tools do not include details on the delays in transfer of the newborn to the NBU and the delays in review of the newborn during admission. |
| **5. Description of clinical illness and progression.** | • A section to comprehensively describe the danger signs at admission and a different section to describe the progression of clinical illness post-admission. | • The WHO child and neonatal death review form include a section to summarise the child's illness and progression in two lines. |
| **6. Nursing care audit.** | • A section to audit the nursing care provided during the hospital stay. | • None of the available audit tools have provisions to audit the nursing care provided to the newborns. |
| **7. Treatment provided.** | • Detailed description of the supportive and definitive treatment provided to the newborn including the feed and fluid management. | • The WHO child and neonatal death review form include a section to describe the treatment provided. This is however not grouped into supportive and definitive treatment. |
| **8. Action plans.** | • A structured action plan summary form that is part of the audit tool. | • The WHO child and neonatal death review form include a structured action plan summary form that is separate from the audit tool.<br>• The other audit tools have a free text section to list down the action plans. |
| **9. Audit meeting attendees.** | • A list with a check box to identify the members of the audit committee who attended the audit meeting. | • None of the available audit tools have a section to document the audit meeting attendees. |
| **10. Modifiable factors.** | • Structured list of modifiable factors categorised into administrative-related, health worker related and patient oriented factors. These have check boxes for selection instead of free text. | • The MPDSR tool has a list with a checkbox for selection. Categorised as the three delays.<br>• The WHO child and neonatal death review form has a list that is categorised as family, administrative and provider related modifiable factors.<br>• The WHO stillbirth and neonatal death case review form has a free text section to list down identified modifiable factors. |

Abbreviations: MPDSR, Maternal and Perinatal Death Surveillance and Response; NBU, Newborn unit; WHO, World Health Organisation.

**Table 2. Components of the audit implementation guide.**

| Component of the newborn audit implementation guide | Standard Operating Procedure |
|---|---|
| 1. Size and composition of the audit committee. | Include all who can influence change.<br>a. Nursing officer-in-charge of NBU and NO in charge of labour ward.<br>b. Senior clinician in NBU (neonatologist/ paediatrician) and Obstetrician/ medical officer from labour ward.<br>c. Hospital administrator–(medical superintendent/hospital administrator/matron in charge of facility).<br>d. Representatives from service departments (Nutrition, pharmacy, laboratory and health records). |
| 2. Roles of the audit committee | Conformity with WHO audit guidelines<br>a. Identifying cases for discussion during the audit meeting.<br>b. Ensuring that records are kept safely and confidentially.<br>c. Providing feedback on audit recommendations to the clinical team and administration.<br>d. Following up on action plans and ensuring they are implemented. |
| 3. Frequency of audit meetings | • Audit meetings to be held two-weekly on a set day and time.<br>• Audit meetings should take 1 hour to 1 hour 30 minutes. |
| 4. How many cases should be audited per session. | • Based on the time allocated to the audit meetings, only one or two cases should be audited per meeting. |
| 5. Criteria for selection of cases for auditing | a. Prevalence (a most common cause of death, increased mortality due to a particular diagnosis).<br>b. Indications that the death is preventable (glaring gaps in the management of a case, preventable diseases or conditions).<br>c. For learning purposes (cases that were difficult to deal with, unexpected deaths, rare cases). |
| 6. Environment during audit meetings | Predictable, all-inclusive and blame-free<br>a. Regular and structured meetings.<br>b. To be held in a spacious room large enough to accommodate all participants.<br>c. Meetings should be all-inclusive.<br>d. Chair of the audit committee should chair the meetings.<br>e. Meetings should be attended by audit participants who can influence change.<br>f. Equality with all participants allowed to express themselves freely.<br>g. Blame-free and non-judgemental environment.<br>h. Environment that maintains confidentiality.<br>i. Should have a strong educational aspect. |
| 7. To ensure the audit cycle is completed. | To ensure action plans are implemented<br>a. Key decision-makers in relevant departments should be made aware of the action plans.<br>b. Direct task allocation and clear role clarification.<br>c. Specific timeframe for implementing what was discussed.<br>d. Taking clear minutes during each meeting and beginning each meeting by reviewing the minutes from the previous meeting.<br>e. Audit team to give regular feedback to hospital administration and hospital management teams on arising recommendations and their implementation status.<br>f. Audit team to follow up on implementation progress with the people tasked to implement them.<br>g. A maximum of three action plans for implementation arising from each audit meeting. |

focus group discussion results were categorised as facilitators or barriers to the audit process. The main themes arising from the FGDs and the participants quotes supporting the findings are provided in S5 Appendix. The themes arising include:

## 1. Perceived Facilitators

i. Patient safety culture which defines an integrated pattern of individual and organizational behaviour that continuously seeks to minimize patient harm that may occur from the care delivery process [46]. The study participants elaborated on how their organisations prioritized patient safety by viewing patient care from a systems perspective. This allowed them to recognize the value of creating an equal environment that supports open dialogue and an environment that encourages learning from preventable adverse events.

ii. Completion of the audit cycle. Poor implementation of audit identified recommendations has been identified as a major contributor to the loss of confidence in the audit process [42,

43, 45]. The study participants described the different strategies that they have put in place to ensure that recommendations are implemented.

## 2. Perceived Barriers

i. Unhealthy organisational culture. This main theme reflects the perceptions of the FGD participants with regard to the impact their organizational culture had on the effectiveness of the audit process.

ii. Knowledge of meaning of clinical audits. Some contributions in the FGDs revealed that some participants understood a clinical audit as simply reviewing the monthly morbidity and mortality statistics focussing on outcomes in the newborn unit. There was no discussion on the processes of care.

iii. Failure to recognise the complexity of the health system and newborn care. The FGD participants revealed that they recognized that newborn care required a team effort. The teams they described were however limited to the immediate newborn care team; nurses and clinicians and occasionally the midwives.

iv. Health workers' perceptions about the value of clinical audits. The FGD participants revealed that the HCW did not fully appreciate the benefits of the clinical audit process on quality improvement.

v. HCW knowledge to perform. The respondents brought out that there was a gap between what the HCW should be doing and what they have the knowledge to do.

**a. Outcome of the consensus meeting to design the implementation guide.** With an understanding of the user requirements through the creation of user personas (S6 Appendix), the team arrived at consensus on the standard operating procedure for conducting the SSNB clinical audit in Kenyan public hospitals using the nominal group technique. Table 2 below describes the seven components of the audit implementation guide and the proposed procedures by which they would be carried out to ensure the completion of each audit cycle.

## The outcome of usability testing of the audit tool and implementation guide

The seven components of the implementation guide were tested in the County hospital to determine if they were feasible for the context. Some of the components remained constant, while others were difficult to implement.

We maintained the four roles of the committee and revised the mandatory members to the six listed below with other members to attend as required:

a. Nursing officer in charge of the newborn unit.

b. Nursing officer in charge of labour ward.

c. Senior most clinician in newborn unit–Neonatologist/paediatrician/medical officer in charge.

d. Obstetrician/ medical officer from labour ward.

e. Nutritionist

f. Hospital administration–medical superintendent/hospital administrator/matron in charge of the facility.

We were successful in holding two-weekly audit meetings except when there were other circumstances beyond our control. Discussing one case took between one and two hours on average. We, therefore, recommended that only one case should be audited per meeting.

To ensure that the action plans were implemented, direct task allocation and giving specific timeframes made it more convenient for the chair of the audit committee to follow up and ensure that they were implemented. We left it as the responsibility of the chair to follow up on the implementation of the action plans either directly or through delegation. The modified implementation guide after the testing period is attached in S7 Appendix.

There were few modifications made to the audit tool during the testing phase and these were mostly based on feedback from the tertiary hospital team. This feedback was more about the content, ensuring that the audit tool was detailed enough for use in the teaching and referral level hospitals. This included restructuring the sections on newborn resuscitation, adding important parameters to the first section on newborn details and widening the scope of options for respiratory support. The final newborn audit tool and implementation guide are attached in S2 and S5 Appendices.

## Discussion

We identified a gap in the availability of a clinical audit tool that comprehensively covers the three periods in the continuum of newborn care. To address this, we used an HCD approach to design a comprehensive SSNB clinical audit tool and its implementation guide. We began by drafting a prototype tool based on identified strengths and weaknesses in the available perinatal, neonatal and child audit tools. Together with HCWs participating in a clinical information network, the prototype tool was further refined and tested to modify design challenges related to its content, factors causing difficult interaction between the user and the tool such as font size, writing space, length of the tool, etc., and the overall user experience. The audit implementation guide contained seven factors that were adapted from the Geneva: World Health Organisation; 2018 manual and modified to build on the strengths and overcome the barriers within the context [2].

### Co-design of the small and sick newborn clinical audit tool

The co-design of the audit tool was centred around the user experiences of the prototypes. The use of real cases in high volume newborn units that managed a diverse range of newborn conditions allowed the users to identify a wide range of modifiable factors at each step in the care process and modify the tool frequently to cater for all these possibilities. While some previous HCD studies have carried out the cognitive walkthrough phase in controlled settings, other studies have used real cases and real scenarios [18, 47]. As expressed by Neyens *et al*., "the use of real cases provides more accurate and detailed information into the experiences and problems that can occur" [47]. The use of real cases however led to several interruptions in the design process due to several HCW strikes and restrictions on meetings due to the COVID-19 pandemic.

The design process was a delicate balance of ensuring that the audit tool was detailed while at the same time being conscious of the human factors, therefore, ensuring effective human-audit tool interaction. The goal was to ensure ease of use to promote adherence. With an understanding of the users who were busy health workers in a constrained environment with other competing interests, there was constant feedback to make the different sections easier and faster to fill. We, therefore, converted the tool that was originally designed on Microsoft

Word into an E-tool on Adobe Acrobat Pro 2020, incorporated checkboxes, textboxes and drop-down calendars, leaving free text-only where necessary. Similar findings have been identified from a study by Muinga *et al.* in the design of a comprehensive newborn monitoring chart where the users preferred the sections of the chart as fixed options to reduce time spent filling it [18]. Other important human factors identified in other studies include screen size, button size, font size, colour, tone and contrast for mobile applications [16, 47].

## Understanding the facilitators and barriers to the audit process and their influence on the design of the audit implementation guide

The extent to which contextual factors influence the success of continuous quality improvement (QI) initiatives such as the clinical audit process cannot be overemphasised. The clinical audit process is a complex intervention whose design must adapt to fit the local context by building on the facilitators and overcoming the barriers to ensure that it accomplishes the desired effects. To understand the contextual factors that informed the design of the components of the implementation guide, we can apply the Informing Quality Improvement Research (InQuIRe) framework developed by Brennan *et al.* to categorise them into the organisational, team and individual-level factors [48].

**Organisational level factors.** In 2016, the perinatal component was included in the maternal death surveillance and response process in Kenya to improve the quality of newborn care in line with the sustainable development goals [49]. The MPDSR guidelines recommend that at the facility level, all maternal deaths should be reviewed within seven days of occurrence, with a set monthly meeting over and above each review meeting [9]. The guideline however recommends that monthly meetings should be convened to review all the perinatal deaths occurring in a facility within the period. The recommendation for multiple maternal death review meetings yet just one monthly perinatal meeting poses a challenge due to the high volume of perinatal and neonatal deaths [20]. The WHO recommends that the frequency of audit meetings should be based on the volume of cases with meetings to be held as frequently as weekly or two-weekly for high mortality areas with an in-depth review of one to two cases [2]. This emphasizes the need to have a separate neonatal clinical audit meeting that complements the MPDSR so that due attention can be placed on the newborn audit process.

The organizational culture has emerged as one that is capable of change evidenced by the support of the clinical audit process through having regular clinical audit meetings and adopting mechanisms to ensure that the audit process leads to change. Despite the presence of an environment that largely advocates for patient safety, the organizational environment is marred with a myriad of challenges that affect the impact of QI initiatives [25, 31, 34, 35, 50]. The emotional fatigue from such an environment and burnout from the high workloads may exacerbate a name and blame culture as identified from the FGDs [51]. Literature that highlights the facilitators and barriers to the audit process proposes mechanisms such as ensuring confidentiality, anonymity, respect and equality among meeting participants and the presence of a multidisciplinary audit committee to promote a no-blame environment [38, 41, 42].

The absence of organizational leadership from the audit meetings has been linked to the poor implementation of action plans arising from the meeting. Filippi *et al.* propose that there is a high likelihood that this reluctance to attend meetings may be due to lack of funds [40]. This may very well apply to the situation in Kenya as with the decentralization of health services, hospitals have lost autonomy to manage and use hospital funds. The result is delayed procurement of essential resources required to maintain a favourable work environment [32]. Therefore, requests are frequently made to the hospital managers which they are unlikely to implement and therefore prefer not to attend QI meetings.

**Team level factors.** Clinical audits are a complex intervention that requires team collaboration to get the desired results. The hierarchical nature of the context has influenced the success of team collaboration in the clinical audit process. As evidenced from the results, the doctors determine if, when and how audit meetings will be conducted with minimal consultation from nursing and other cadres. The audit meetings have therefore been viewed as a doctors' affair where the discussion will be dominated by the doctors and the focus will be on the medical aspects of care. Constituting an interdisciplinary team to manage the audits would be a solution to this problem [43]. In complex adaptive systems, each component must understand how important its contribution is to the whole [52]. This informed the selection of an interdisciplinary team whose actions are tightly linked and would influence the quality of newborn care. Successful clinical audit teams include those with well-defined roles for each member to prevent free riding whereby some members fail to contribute their fair share in team effort [53, 54]. Other characteristics of successful teams have been identified as those that have a committed chair who supports QI initiatives, are diverse and constitute the key decision-makers from departments involved in newborn care, have hospital managers as part of the team, promotes equality, open dialogue and confidentiality during the audit meetings [40, 43, 45].

**Individual-level factors.** The poor implementation of audit recommendations has been widely recognized as a deterrent to the success of the audit process [42, 55]. Though this can be attributed to poor support from leadership, another important factor that plays into this is the poor skills in problem identification. As identified in the literature, a large proportion of the problems in newborn care in LMICs can be solved through the appropriate use of available resources and by improving knowledge and skills of the frontline health workers [56, 57]. These 'simple' measures added up play a significant role in improving the quality of newborn care as the health workers await implementation of the more resource-intensive solutions.

## Strengths and limitations

The main strength of the design process was the multiple iterations involving the end-users. In addition, we had senior representatives in the field of newborn medicine actively participating in the design process and this contributed to the development of a comprehensive small and sick newborn clinical audit process. The use of real cases during the iteration process enabled the design of an audit tool that could adequately capture the modifiable factors in the context.

The limitations of the study included the fact that before the focus group discussions, we shared the Geneva: World Health Organisation; 2018 manual with the participants [2]. The reason for this was to allow them to gain some knowledge on the recommended standards for conducting a clinical audit to enrich the discussions. This however resulted in the contamination of the FGDs as the participant responses seemed to be what was expected, but not necessarily what happened. Another limitation was the inability to include more than two hospitals in the testing phase. The intention was to include as many hospitals as possible, however, this was disrupted by the COVID-19 pandemic and a nationwide strike that involved most of the hospitals except for the two that were included. The two hospitals represented a high-volume county referral hospital and a tertiary level hospital. This may therefore not be generalisable to lower level hospitals which may not have the capacity to implement the implementation guide as designed.

## Conclusion

The use of a Human-Centred Design process enabled the design team and the users to design a high-quality audit tool and implementation guide that can achieve its intended goals with

efficiency, effectiveness and satisfaction while considering the capabilities and limitations of the end-users within their context.

## Supporting information

**S1 Data. Supporting data files.** Including the audit prototype tools for each stage of the cognitive walkthrough, the consensus workshop data, and the data from the focus group discussions.
(RAR)

**S1 Appendix. Description of consensus workshop to develop a SSNB audit implementation guide.**
(DOCX)

**S2 Appendix. Consent form template for focus group discussions.**
(PDF)

**S3 Appendix. Draft zero and final high-quality prototype of NBU audit tools.**
(RAR)

**S4 Appendix. Categorisation of identified changes to the prototype audit tool based on usability, human factors and user experience and the outcomes of the feedback.**
(DOCX)

**S5 Appendix. Focus group discussion results.**
(DOCX)

**S6 Appendix. User personas.**
(PDF)

**S7 Appendix. Modified implementation guide.**
(RAR)

## Acknowledgments

We would like to acknowledge The Initiative to Develop African Research Leaders (IDeAL) for the financial and technical support to conduct this research. We would also like to thank Beth Maina and the entire newborn care staff at Pumwani Maternity Hospital and Kenyatta National Hospital for taking part in the audit meetings and providing feedback on the design of the tool. The paediatric residents (Allan Kayiza, Joy Odhiambo, Rachel Kanguha and Tahniya Jhuthi), Carolyne Malingu of Kenyatta National Hospital and Peris Musitia, Daniel Mbuthia, Conrad Wanyama and Livingstone Mumelo of KEMRI Wellcome Trust Research Programme for assisting with the moderation of the design workshop.

## Author Contributions

**Conceptualization:** Muthoni Ogola, Naomi Muinga, Jalemba Aluvaala, Mike English, Grace Irimu.

**Data curation:** Muthoni Ogola, Jalemba Aluvaala, Mike English, Grace Irimu.

**Formal analysis:** Muthoni Ogola, Naomi Muinga, Jalemba Aluvaala, Mike English, Grace Irimu.

**Funding acquisition:** Mike English.

**Investigation:** Muthoni Ogola, John Wainaina, Maureen Muriithi, Jalemba Aluvaala, Mike English, Grace Irimu.

**Methodology:** Muthoni Ogola, John Wainaina, Jalemba Aluvaala, Mike English, Grace Irimu.

**Project administration:** Muthoni Ogola, Jalemba Aluvaala, Mike English, Grace Irimu.

**Resources:** Muthoni Ogola, John Wainaina, Naomi Muinga, Wairimu Kimani, Maureen Muriithi, Jalemba Aluvaala, Mike English, Grace Irimu.

**Software:** Muthoni Ogola, John Wainaina, Naomi Muinga, Jalemba Aluvaala, Mike English.

**Supervision:** Muthoni Ogola, Wairimu Kimani, Maureen Muriithi, Jalemba Aluvaala, Mike English, Grace Irimu.

**Validation:** Muthoni Ogola, Jalemba Aluvaala, Mike English, Grace Irimu.

**Visualization:** Muthoni Ogola, Naomi Muinga, Wairimu Kimani, Jalemba Aluvaala, Mike English, Grace Irimu.

**Writing – original draft:** Muthoni Ogola.

**Writing – review & editing:** Muthoni Ogola, Jalemba Aluvaala, Mike English, Grace Irimu.

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
