## [Decision Letter · Decision Letter 0]

11 Nov 2022

PGPH-D-22-00756

Development of a small and sick newborn clinical audit tool and its implementation guide using a human-centred design approach Newborn clinical audit process and design

Dear Dr. Ogola,

Thank you for submitting your manuscript to PLOS Global Public Health. After careful consideration, we feel that it has merit but does not fully meet PLOS Global Public Health’s publication criteria as it currently stands. Therefore, we invite you to submit a revised version of the manuscript that addresses the points raised during the review process.

We look forward to receiving your revised manuscript.

Kind regards,

Henry Zakumumpa, PhD

Academic Editor

Journal Requirements:

1. We have noticed that you have uploaded "Supporting Data" in Supporting Information files, but you have not included a list of legends. Please add a full list of legends for your Supporting Information files after the references list. 

Additional Editor Comments (if provided):

We are delighted to share comments from our reviews. Please include a point by point response to each of the comments raised so we can swiftly move to a decision.

Both reviews have touched on the focus group discussion component in this manuscript in terms of methods but also in the their representation in the results please attend to their concerns.

Secondly, the reviews suggest a need to make the reporting of methods and approaches more concise and efficient. One of the reviewers has taken the extra burden to pin point specific parts of the methods that need a re-write.

I look forward to reading a revised manuscript.

Reviewers' comments:

Reviewer's Responses to Questions

**Comments to the Author**

1. Does this manuscript meet PLOS Global Public Health’s publication criteria? Is the manuscript technically sound, and do the data support the conclusions? The manuscript must describe methodologically and ethically rigorous research with conclusions that are appropriately drawn based on the data presented.

Reviewer #1: Partly

Reviewer #2: Yes

2. Has the statistical analysis been performed appropriately and rigorously?

Reviewer #1: N/A

Reviewer #2: N/A

3. Have the authors made all data underlying the findings in their manuscript fully available (please refer to the Data Availability Statement at the start of the manuscript PDF file)?

Reviewer #1: Yes

Reviewer #2: Yes

4. Is the manuscript presented in an intelligible fashion and written in standard English?

Reviewer #1: Yes

Reviewer #2: Yes

5. Review Comments to the Author

Reviewer #1: This is a well written article addressing an important issue for newborn health and providing a thoughtful approach to development of a clinical audit tool.

Some comments for consideration by the authors:

The part of the methods section describing study setting / participants is very long and would benefit from sub-headings to sign-post/guide the reader, linked to the phases of the study. The description of study sites is very long and the article would benefit from this section being more concise.

Box 2 – It would be helpful to have the numbers of each participant type in the box, for ease of access by readers

In general, the methods are very long and contain very detailed and extraneous information which could be removed, to make the manuscript more concise and readable:

E.g. detailing the day of the week when meetings were held is not necessary, instead it is sufficient to state that meetings were initially held monthly then every two-weeks (lines 174 – 177). Detailing who prepared the meetings is not necessary in the text, as this is outlined in Box 3. I suggest the authors review the methods section and edit the word count down where possible, using clearly labelled sub-headings.

This is a complicated mixed methods study with multiple components occurring at different timepoints. A figure giving an overview of the activities and timelines would be very helpful for the reader to appreciate how long it took and how the different components overlapped temporally.

The authors state that a focus group discussion was conducted. More information about this qualitative method is required - particularly who conducted the FGD, what was the intended sample size for the FGD and how were participants selected, how was the data analysed, any methods used to enhance the trustworthiness & reliability of the data.

Line 232 – verbal or written consent was obtained prior to the FGDs – this is slightly unusual. In which cases were verbal consent obtained and why not always written consent?

There are some aspects of the results section which are more relevant to the methods section. E.g. Line 264, lines 286 – 289, 307 - 308. I suggest focus the methods section on the process needed to generate the audit tool / implementation guide and the results section on the actual tool/guide. Also, use of sub-headings to sign post readers to particular findings would be beneficial, as it is hard to navigate the results section and come away with clear understanding of what was found.

The FGD findings are presented half way through the results section and may be lost to the reader. I would suggest that the authors consider making these findings more prominent in the manuscript, as there is rich learning from this part of the study around facilitators / barriers.

Table 2 contains useful, practical information for other centres wanting to do similar work.

Lines 354 – 358: This level of detail about some members of the audit committee not attending all meetings is not required. Consider making this paragraph more concise and focus on the roles of the committee members instead.

Lines 366 – 387: It is not clear to me how this information is a finding of the study? This may be better suited to being in the discussion section

The first paragraph of the discussion section is well written and summarises the findings / work nicely. However, the rest of the discussion is very long and would benefit from being shorter and focusing only on key findings and learning with clear sub-headings relating to the objectives of the study.

In summary, this manuscript has the potential to add to the literature for this topic but requires more work to clearly present the methods and findings in a coherent and concise way. In addition, more information about the methods of the qualitative aspect (FGDs) is required.

Reviewer #2: Thank you very much for an opportunity I had to review this interesting paper. Well

done to authors. The paper is well written and it’s introducing a novel approach to use

during clinical audit for neonates.

But I have few suggestions for authors to consider.

1. Abstract is well written but line 31, the authors have used abbreviation (NBUs) without defining it. I suggest writing NBUs in full in the abstract.

2. Introduction is also well written, but to make it clearer, few suggestions should be considered.

a) -Line 40, majority of deaths-add the figures to quantify the burden

- Line-69-70. You have introduced HCD well and mentioned how it is used,

b) Please elaborate more on what is HCD approach, and rationale for its use in clinical audits.

3. Methods section

a) Line 117- What criteria/s did you use to choose PMH to test usability of the

audit tool?

b) It’s not clear if the tool was meant for auditing deaths, or neonatal cases or both, please clarify

c) You used consensus meetings and FGDs as some of methodologies- elaborate more on who was involved in FGDs, sampling and how you analysed FGD data

4. Results

a) Table one- good way of showing what is available on the ground but it could be better if the new tool developed in this study could be also compared against what is currently available to show uniqueness of the tool developed in this study in Table 1

b) When describing the outcome of available tools on line 248-251, you have described about gaps with Kenya MPDSR tool, WHO stillbirth and neonatal tool and paediatric audit tool. There is mention of perinatal as well as neonates. Definition of perinatal, neonates and its cut off age would assist to make clear where gap is interns of tools that are already in existence and the one you have developed.

c) On FGDs- you have not included participants quotes to support your summary of findings. I have seen quotes in supporting information appendix 5. Include a statement for readers to refer to participants quotes in supplementary information when reading the summary of FGDs findings

5. Strengths and limitation

a) As indicated you only included 2 hospitals- How are the findings applicable to other facilities? Would you please clarify generalisability of findings in your limitation/strength section

6. PLOS authors have the option to publish the peer review history of their article (what does this mean?). If published, this will include your full peer review and any attached files.

**Do you want your identity to be public for this peer review?** For information about this choice, including consent withdrawal, please see our Privacy Policy.

Reviewer #1: **Yes: **Helen Brotherton

Reviewer #2: No

---

## [Decision Letter · Decision Letter 1]

18 Jan 2023

Development of a small and sick newborn clinical audit tool and its implementation guide using a human-centred design approach Newborn clinical audit process and design

PGPH-D-22-00756R1

Dear Dr Muthoni Ogola,

We are pleased to inform you that your manuscript 'Development of a small and sick newborn clinical audit tool and its implementation guide using a human-centred design approach Newborn clinical audit process and design' has been provisionally accepted for publication in PLOS Global Public Health.

Best regards,

Henry Zakumumpa, PhD

Academic Editor

Thank you for attending to the comments raised by our reviewers. We are delighted to accept this paper.

Reviewer Comments (if any, and for reference):

Reviewer's Responses to Questions

**Comments to the Author**

1. If the authors have adequately addressed your comments raised in a previous round of review and you feel that this manuscript is now acceptable for publication, you may indicate that here to bypass the “Comments to the Author” section, enter your conflict of interest statement in the “Confidential to Editor” section, and submit your "Accept" recommendation.

Reviewer #1: All comments have been addressed

Reviewer #2: All comments have been addressed

2. Does this manuscript meet PLOS Global Public Health’s publication criteria? Is the manuscript technically sound, and do the data support the conclusions? The manuscript must describe methodologically and ethically rigorous research with conclusions that are appropriately drawn based on the data presented.

Reviewer #1: Yes

Reviewer #2: Yes

3. Has the statistical analysis been performed appropriately and rigorously?

Reviewer #1: Yes

Reviewer #2: Yes

4. Have the authors made all data underlying the findings in their manuscript fully available (please refer to the Data Availability Statement at the start of the manuscript PDF file)?

Reviewer #1: Yes

Reviewer #2: Yes

5. Is the manuscript presented in an intelligible fashion and written in standard English?

Reviewer #1: Yes

Reviewer #2: Yes

6. Review Comments to the Author

Reviewer #1: (No Response)

Reviewer #2: Well done to all authors for responding to issues raised. I am happy with responses.The issues raised have been addressed accordingly and improved the paper. All the best!

7. PLOS authors have the option to publish the peer review history of their article (what does this mean?). If published, this will include your full peer review and any attached files.

**Do you want your identity to be public for this peer review?** For information about this choice, including consent withdrawal, please see our Privacy Policy.

Reviewer #1: No

Reviewer #2: **Yes: **Dr Mtisunge Joshua Gondwe
